# Measuring the State of Document Understanding

**Łukasz Borchmann**∗         **Michał Pietruszka**∗         **Tomasz Stanisławek**∗

**Dawid Jurkiewicz**     **Michał Turski**     **Karolina Szyndler**     **Filip Graliński**

Applica.ai
`firstname.surname@applica.ai`

## Abstract

Understanding documents with rich-layouts plays a vital role in digitization and hyper-automation but remains a challenging topic in the NLP research community. Additionally, the lack of a commonly accepted benchmark made it difficult to quantify progress in the domain. To empower research in Document Understanding, we present a suite of tasks that fulfill the highest quality, difficulty, and licensing criteria. The benchmark includes Visual Question Answering, Key Information Extraction, and Machine Reading Comprehension tasks over various document domains, and layouts featuring tables, graphs, lists, and infographics. The current study reports systematic baselines making use of recent advances in layout-aware language modeling. To support adoption by other researchers, both the benchmarks and reference implementations will be shortly released.

## 1   Introduction

While mainstream Natural Language Processing focuses on plain text documents, content one encounters when reading, e.g., scientific articles, company announcements, or even personal notes, is rarely plain and purely sequential. In particular, the document's visual and layout aspects that guide our reading process and carry non-textual information appear to be an essential aspect that requires comprehension. These layout aspects, as we understand them, are prevalent in tasks that can be much better solved when given not only sequence text on the input but pieces of multimodal information covering aspects such as text-positioning (i.e., location of words on the 2D plane), text-formatting (e.g., different font sizes, colors), and graphical elements (e.g., lines, bars, presence of figure) among others.

Over the decades, systems dealing with document understanding developed an inherent aspect of multi-modality that nowadays revolves around the problems of integrating visual information with spatial relationships and text [34, 1, 49, 11]. Within this frame of reference, Document Understanding may involve the ability to comprehend documents by integrating information from different modalities, e.g., to analyze the figure in the context of accompanying text [2].

In general, when document processing systems are considered, the term *understanding* is thought of specifically as the capacity to convert a document into meaningful information [9, 56, 14]. The exact nature of this information depends on the task under consideration and can range from the location of document components to the answer valid for some content-related questions formulated in natural language.

Despite its importance for digital transformation, the problem of measuring how well available models obtain information from a wide range of document types and how suitable they are for

---

∗Equal contribution

Submitted to the 35th Conference on Neural Information Processing Systems (NeurIPS 2021) Track on Datasets and Benchmarks. Do not distribute.

freeing workers from paperwork through process automation is not yet addressed. We intend to bridge this major gap by introducing the first Document Understanding benchmark (see Section 5 for a review). It includes tasks that either originally had a vital layout understanding component or were reformulated in such a way that after our modification requires layout understanding. In particular, there is no structured representation of the underlying text, such as a database-like table given in advance, and it has to be determined as a part of the end-to-end process from the raw input file. Every time, there is only a PDF file provided as an input with accompanying textual tokens and their locations (bounding boxes). It is not enough to process the text in a sequential manner (token by token), and there is no ground truth reading order given in advance. There are also some common document understanding problems involved (Section 1.2).

**Contribution.** We review and evaluate available data to asses its quality, provide manually annotated diagnostic sets, measure the human performance, improve data splits, and correct the existing manual annotations. Importantly, part of the existing datasets is reformulated in a document understanding paradigm, such as a more competitive problem fitting real-world situations is derived. Additionally, we propose a novel format for storing data, and provide datasets in an unified form, making their joint processing and evaluation more accessible. Finally, we provide and open source baselines solving the task, to facilitate further research on the problem.

## 1.1 Importance and applications

As a means of end-to-end process automation, Document Understanding fits into the rapidly growing market of hyperautomation-enabling technologies, estimated to reach nearly $600 billion in 2022, up 24% from 2020 [40]. One of the core bottlenecks for this growth is a demand for structured data that serves technologies such as, e.g., big data analytics and workflow automation. Considering that unstructured data is orders of magnitude more abundant than structured data, the lack of necessary tools to extract and analyze it can limit the performance of these intelligent services. The process of structuring data and content must be robust to various document domains. It should also not assume a static or template layout since the diversity of documents and formats is increasing. A good document understanding *benchmark* should measure to what extent these technologies are supported in their tasks.

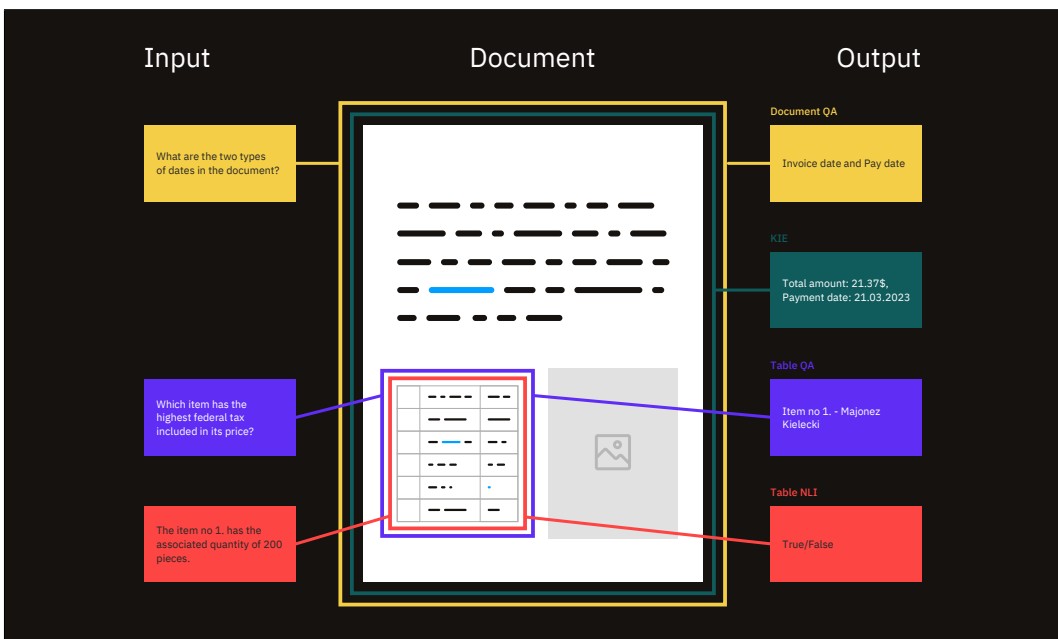

Figure 1: Document Understanding covers problems ranging from the extraction of key information, through verification statements related to rich content, to answering open questions regarding an entire file. It may involve the comprehension of multi-modal information conveyed by a document.

## 1.2 Challenges

Owing to its end-to-end nature and heterogeneity, Document Understanding is the touchstone of Machine Learning. The challenges begin to pile up due to the mere form a document is available in, as there is a widespread presence of analog materials such as scanned paper records.

**C1.** Consequently, architectures evaluated with different OCR engines are incomparable, e.g., it has been shown that the choice of an OCR engine may impact results more than the choice of model architecture [41]. The overall performance is affected by the noise resulting from OCR errors and incorrectly detected reading order, which impacts commonly used solutions based on sequence labeling. The latter problem is currently being investigated with models independent of sequential order [19, 38].

**C2.** Yet another layout-related problem can be exemplified by trying to understand a value from a table cell. In Document Understanding, contrary to the problems of QA over tables, there is no parsed table given in advance as both born-digital and analog documents lack such information. As a result, the application of a particular architecture might require layout analysis and inference of table structure as a necessary step. Nevertheless, it is common to rely on end-to-end models comprehending spatial relationships between the bounding boxes of words instead [63, 55, 38].

**C3.** In addition to layout and textual semantics, part of the covered problems demand a Computer Vision component, e.g., to detect a logo, analyze a figure, recognize text style, determine whether the document was signed or the checkbox nearby was selected. Thus, Document Understanding naturally incorporates challenges of both multi-modality and each modality individually.

**C4.** Moreover, it is common that token-level annotation is not available, and one receives merely key-value or question-answer pairs assigned to the document. Even in problems of extractive nature, token spans cannot be easily obtained, and consequently, the application of state-of-the-art architectures from other tasks is not straightforward. In particular, authors attempting Document Understanding problems in sequence labeling paradigms were forced to rely on faulty handcrafted heuristics [38].

While part of the mentioned challenges are either task- or dataset-specific, they are widespread across Document Understanding problems. A good Document Understanding model should achieve high accuracy and work robustly for documents with the challenges mentioned above.

## 1.3 Desiderata

**Gather.** We define our desiderata as follows. We wish to gather both the sparse Document Understanding datasets published over the years and datasets from related fields that can be reformulated for Document Understanding. Even though a plethora of commercial solutions deal with the problem and it is an object of increasing interest, the availability of public datasets in this field is limited [42]. It results from the common practice of publishing works with an evaluation performed on a private, presumably confidential dataset.

**Examine.** Then, we intend to examine the value of gathered resources, select the most promising, eliminate their identified disadvantages and provide missing information wherever applicable. Our changes may include improvements of annotation quality and dataset splits, elimination of biases, or preparation of human baselines.

**Unify.** To eliminate some of the barriers in future experiments, we wish to propose a format to unify varied Document Understanding tasks and convert all of the datasets included in the benchmark. Additionally, to address challenge C1 (Section 1.2), we provide versioned OCR layers for scanned documents to make models evaluated in the future directly comparable.

**Evaluate.** To show there is much space for improvement, we intend to evaluate state-of-the-art models and comment on their result comparisons and human baselines. By precisely diagnosing aspects where these models underperform, we wish to aid the community in identifying where to focus their efforts to conduct valuable research and development.

**Open.** Our stance on the future of the Document Understanding Benchmark is to be open and evolving. With further deep learning advances, some tasks may be considered solved, and benchmarks need to hold to that pace. Given the scarcity of datasets available that conform to our Design Process criteria of quality, difficulty, and licensing (see Section 3.1 for detailed analysis), we intend to mimic

changes in the publicly available datasets. Specifically, we view an extension of our suite as a continuous process when there is a new dataset complying with defined standards.

## 2 Landscape of Document Understanding tasks

For the purposes of the present work, we treat Document Understanding as an umbrella term covering problems of Key Information Extraction, Classification, Question Answering, Layout Analysis, and Machine Reading Comprehension whenever they involve rich documents in contrast to plain-text or image-text pairs (Figure 1).

In addition to the problems strictly classified as Document Understanding, several related tasks can be reformulated as such. These provide either text-figure pairs instead of real-world documents or parsed tables given in their structured form. Since both can be rendered as synthetic documents with some loss of information involved, they are worth considering bearing in mind the low availability of proper Document Understanding tasks. Importantly, such reformulated tasks share an important aspect of C2 challenge we outlined in Section 1.2, as content interpretation is no longer available.

**KIE.** Key Information Extraction, also referred to as Property Extraction, is a task where (properties, document) tuple values are to be provided. Contrary to QA problems, there is no question in natural language but rather a phrase or keyword, such as *total amount*, or *place of birth*. Public datasets in the field include extraction performed on receipts [17, 35], invoices, reports [41], and forms [21]. Documents within each of the mentioned tasks are homogeneous, whereas the set of properties to extract is limited and known in advance – in particular, the same type-specific property names appear in both test and train sets. In contrast to Name Entity Recognition, KIE typically does not assume token-level annotations are available, and may require to normalize values found within the document. Moreover, accurate prediction of property values requires some form of layout comprehension.

**QA and MRC.** At first glance, Question Answering and Machine Reading Comprehension over Documents is simply the KIE scenario where a question in natural language replaced a property name. More differences become evident when one notices that QA and MRC involve an open set of questions and various document types. Consequently, there is pressure to interpret the question and to possess better generalization abilities. Furthermore, a specific content to analyze demands a much stronger comprehension of visual aspects, as the questions commonly relate to figures and graphics accompanying the formatted text [29, 28, 46].

**Classification.** Though document image classification was initially approached using solely the methods of Computer Vision, it has recently become evident that multi-modal models can achieve significantly higher accuracy [54, 55, 38]. Similar conclusions were recently reached in other tasks, e.g., assigning labels to excerpts from biomedical papers depending on the used experiment method [53]. Classification in our context involves rich content, where comprehension of both visual and textual aspects is required since unimodal models underperform.

**Layout analysis.** Document Layout Analysis, performed to determine a document's components, is the oldest Document Understanding problem, initially motivated by the need to optimize storage and the transmission of large information volumes [34]. Even though the motivation behind it has changed over the years, it is rarely an end itself but rather a means to achieve a different goal, such as improving OCR systems. A typical dataset in the field assumes detection and classification of page regions or tokens, depending on the area they belong to [62, 26].

**QA over figures.** Question Answering over Figures is, to some extent, comparable with QA and MRC over documents described above. The difference is that a 'document' here consists of a single born-digital plot, reflecting information from chosen, desirably real-world data. Because questions are typically templated and figures generated by authors of the task, regular datasets in this category contain millions of examples [31, 4]. Interestingly, questions here can be demanding, e.g., require the estimation of a line chart value at some point.

**QA and NLI over tables.** Question Answering and Natural Language Inference over Tables are similar, though in the case of NLI, there is a statement to verify instead of a question to answer. There is never a need to analyze the actual layout, as both assume comprehension of a provided data structure in a way that is equivalent to a database table. Consequently, the methods proposed here are distinct from those used in Document Understanding. There are, however, similarities to exploit, i.e., every task of this type can be reformulated as Document Understanding by simply rendering

the table and treating it as an actual document. Apart from the NLI specific, the resulting scenario is similar to QA and MRC over documents described earlier, with the difference that a 'document' now consists of a single born-digital table.

## 3 Benchmark overview

Many of the datasets existing in the previously analyzed landscape cannot, on their own, provide enough information that would allow scientists to generalize results to other tasks within the document understanding. Here we describe the proposed suite of tasks and the designed process that led to its creation. Extensive documentation of the process, including the datasheet, is available in Appendices A-H and supplementary materials.

### 3.1 Desired characteristics

As the value and importance of Document Understanding result from its application to process automation, a good benchmark should measure to which degree workers can be supported in their tasks. Though Layout Analysis is oldest of the Document Understanding problems, its output is often not an end in itself but rather a half-measure disconnected from the final information the system is used for. Consequently, we excluded all datasets of this kind by design and restricted ourselves to English tasks of classification, KIE, QA, MRC, and NLI over complex documents, figures, and tables.

Candidate tasks resulted from an extensive review of both literature and data science challenges without accompanying publication. Gathered proposals were filtered according to the criteria of quality, difficulty, and licensing.

**Quality.** Availability of high-quality annotation was a condition *sine qua non* for a task to qualify. To ensure the highest annotation quality, we excluded resources prepared using a distant annotation procedure, e.g., classification tasks where entire sources were labeled instead of individual instances, or templated question-answer pairs.

**Difficulty.** As it makes no sense to measure progress on solved problems, only tasks with a substantial gap between human performance and state-of-the-art models were considered. In the case of promising tasks lacking a human baseline, we provided our estimation.

**Licensing.** In publishing our benchmark, we are making efforts to ensure the highest standards for the future of the machine learning community. Only tasks with a permissive license to use annotations and data for further research can be considered.

At the same time, we recognized it is essential to approach the benchmark construction holistically, i.e., to carefully select tasks from diverse domains and types in the rare cases where datasets are abundant.

### 3.2 Selected tasks

Table 1 summarizes the selected tasks described in detail below, whereas Appendix G covers the complete list of considered datasets and reasons we omitted them. Lack of the classification and figure QA tasks in this selection results from the fact that none of the available fulfills the assumed selection criteria.

The ★ symbol denotes that the dataset was reformulated or modified to improve its quality or align with the Document Understanding paradigm (See Table 2 and Appendix B). We do not distinguish with this mark minor changes, such as data deduplication introduced in multiple datasets (Appendix A).

**DocVQA.** Dataset for Question Answering over single-page excerpts from various real-world industry documents. Typical questions present here might require comprehension of images, free text, tables, lists, forms, or their combination [29]. The best-performing solutions so far make use of layout-aware multi-modal models employing either encoder-decoder or sequence labeling architectures [38, 55]. We take the dataset as is without introducing any modification.

**InfographicsVQA.** The task of answering questions about visualized data from a diverse collection of infographics, where the information needed to answer a question may be conveyed by text, plots, graphical or layout elements. Currently, the best result is obtained by an encoder-decoder model.

**Kleister Charity.**  A task for extracting information about charity organizations from their published reports is considered, as it is characterized by careful manual annotation by linguists and a significant gap to human performance. It addresses important areas, namely high layout variability (lack of templates), need for performing an OCR, the appearance of long documents, and multiple spatial features (e.g., tables, lists, and titles). We take the dataset as is without introducing any modification.

**PWC★.**  Papers with Code Leaderboards dataset was designed to extract result tuples from machine learning papers, including information on task, dataset, metric name, score. The best performing approach involves a multi-step pipeline, with modules trained separately on identified subproblems [24]. In contrast to the original formulation, we provide a complete paper as input instead of the table. This approach allows us to treat the problem as an end-to-end Key Information Extraction task with grouped variables (Appendix B).

**DeepForm★.**  KIE dataset consisting of socially important documents related to election spending. The task is to extract contract number, advertiser name, amount paid, and air dates from advertising disclosure forms submitted to the Federal Communications Commission [44]. We use a subset of distributed datasets and improve annotations errors (Appendix B).

**WikiTableQuestions★.**  Dataset for QA over semi-structured HTML tables sourced from Wikipedia. The authors intended to provide complex questions, demanding multi-step reasoning on a series of entries in the given table, including comparison and arithmetic operations [36]. The problem is commonly approached assuming a semantic parsing paradigm, with an intermediate state of formal meaning representation, e.g., inferred query or predicted operand to apply on selected cells [59, 16]. We reformulate the task as document QA by rendering the original HTML and restrict available information to layout given by visible lines and token positions (Appendix B). It is forbidden for participating systems to use the HTML source.

**TabFact★.**  To study fact verification with semi-structured evidence over relatively clean and simple tables collected from Wikipedia, entailed and refuted statements corresponding to a single row or cell were prepared by the authors of TabFact [6]. Without being affected by the simplicity of binary classification, this task poses challenges due to the complex linguistic and symbolic reasoning required to perform with high accuracy. Analogously to WikiTableQuestion, we render tables and reformulate the task as document NLI (Appendix B).

Challenges we outlined in Section 1.2 are prevalent in this selection. In particular, scanned documents and infographics have a crucial component of OCR-related problems (C1), no task provides easily accessible information on content interpretation (C2) or token-level annotations (C4), and they may demand comprehension of visual clues to perform well (C3).

### 3.3 Diagnostic subsets

We propose several auxiliary validation subsets, spanning across all the tasks, to improve result analysis and aid the community in identifying where to focus its efforts. A detailed description of these categories and related annotation procedures is provided in Appendix E.

**Answer characteristic.**  We consider four features regarding the answer shallow characteristic. First, we indicate whether the answer is provided in the text explicitly in exact form (*extractive* data point)

| Task | Size (thousands) | | | Type | Metric | Features | | Domain |
| --- | --- | --- | --- | --- | --- | --- | --- | --- |
| | Train | Dev | Test | | | Input | Scanned | |
| DocVQA | 10.2 | 1.3 | 1.3 | Visual QA | ANLS | | + | Business |
| InfographicsVQA | 4.4 | .5 | .6 | Visual QA | ANLS | | − | Open |
| Kleister Charity | 1.7 | .4 | .6 | KIE | F1 | Doc. | +/− | Legal |
| PWC★ | .2 | .06 | .12 | KIE* | F1 | | − | Scientific |
| DeepForm★ | .7 | .1 | .3 | KIE | F1 | | +/− | Finances |
| WikiTableQuestions★ | 1.4 | .3 | .4 | Table QA | Acc. | Exc. | − | Open |
| TabFact★ | 13.2 | 1.7 | 1.7 | Table NLI | Acc. | | − | Open |

Table 1: Comparison of selected tasks with their base characteristic, including information regarding whether an input is an entire document (Doc.) or document excerpt (Exc.)

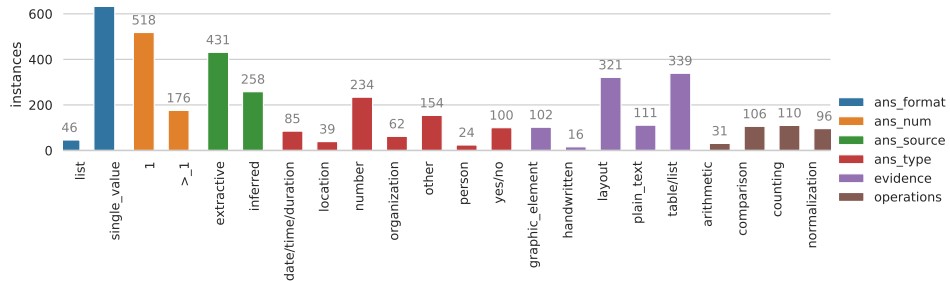

Figure 2: Number of annotated instances in each diagnostic subset category.

or has to be inferred from the document content (*abstractive* one). The second category includes, e.g., all the cases where value requires normalization before being returned (e.g., changing the date format). Next, we distinguish expected answers depending on whether they contain a *single value* or *list* of values. Finally, we decided to recognize several popular data types depending on shapes or class of expected named entity, i.e., to distinguish *date, number, yes/no, organization, location, and person* classes.

**Evidence form.** As we intend to analyze systems dealing with rich data, it is natural to study the performance w.r.t. the form that evidence is presented within the analyzed document. We distinguished *table/list, plain text, graphic element, layout,* and *handwritten* categories.

**Required operation.** Finally, we distinguish whether i.e., *arithmetic operation, counting, normalization* or some form of *comparison* has to be performed to answer correctly.

## 3.4 Unified format

We propose a unified format for storing information in the Document Understanding domain and deliver converted datasets as part of the released benchmark. It assumes three interconnected dataset, document annotation and document content levels. Please refer to the repository for examples and formal specifications of the schemes.

**Dataset.** The dataset level is intended for storing the general metadata, e.g., name, version, license, and source. Here, the JSON-LD format based on the well-known schema.org web standard is used.[2]

**Document.** The documents annotation level is intended to store annotations available for individual documents within datasets and related metadata (e.g., external identifiers). Our format, valid for all of the Document Understanding tasks, is specified using the JSON-Schema standard. This ensures that every record is well-documented and makes automatic validation possible. Additionally, to make the processing of large datasets efficient, we provide JSON Lines file for each split, thus it is possible to read one record at a time.

---

[2]See `https://json-ld.org/` for information on the JSON-LD standard, and `https://developers.g oogle.com/search/docs/data-types/dataset` for the description of adapted schema.

| Dataset | Diagnostic sets | Unified format | Human performance | Manual annotation | Reformulation as DU | Improved split |
|---|---|---|---|---|---|---|
| DocVQA | + | + | − | − | − | − |
| InfographicsVQA | + | + | − | − | − | − |
| Kleister Charity | + | + | − | − | − | − |
| PWC | + | + | + | + | + | − |
| DeepForm | + | + | + | + | − | + |
| WikiTableQuestions | + | + | + | − | + | + |
| TabFact | + | + | − | − | + | − |

Table 2: Brief characteristics of our contribution, major fixes and modifications introduced to particular datasets. See Appendix B for a full description.

**Content.** As part of the original annotation or additional data we provide is related to document content (e.g., the output of a particular OCR engine that is of critical importance due to C2), we introduce the document's content level. Similarly to the document level, we propose an adequate JSON Schema and provide the JSON Lines files in addition. PDF files with the source document accompany dataset -, document-, and content-level annotations. If the source PDF was not available, a lossless conversion was performed.

## 3.5 Human performance

Estimation of human performance for PWC, WikiTableQuestions, DeepForm was performed in-house by professional annotators who are full-time employees of our company after completing the task-specific training (See Appendix D). Each dataset was approached with two annotators; the average of their scores, when validated against the gold standard, is treated as the human performance (See Table 3). Interestingly, human scores on PWC are relatively low in terms of F1 value – we explained this and justified keeping the task in Appendix B.

## 3.6 Evaluation protocol

All the benchmark submissions are expected to conform to the following rules to guarantee fair comparison, reproducibility, and transparency.

1. All results should be automatically obtainable starting from either raw PDF documents or the JSON files we provide. In particular, it is not permitted to rely on the potentially available source file that our PDFs were generated from or in-house manual annotation.

2. Despite the fact that we provide an output of various OCR mechanisms wherever applicable, it is allowed to use software from outside the list. In such cases, participants are highly encouraged to donate OCR results to the community, and we declare to host them along with other variants. It is expected to provide detailed information on used software and its version.

3. Any dataset can be used for unsupervised pretraining. The use of supervised pretraining is limited to datasets where there is no risk of information leakage, e.g., one cannot train models on datasets constructed from Wikipedia tables unless it is guaranteed that the same data does not appear in WikiTableQuestions and TabFact.

4. It is encouraged to either use datasets already publicly available or to release private data used for pretraining. The minimum requirement for a private dataset is the description of its size and creation sufficient to reproduce the results.

5. Training performed on a development set is not allowed. We assume participants select the model to submit using training loss or validation score. We do not release test sets and keep them secret by introducing a daily limit of evaluations performed on the benchmark's website.

6. Although we allow submissions limited to one category, e.g., QA or KIE, complete evaluations of models that are able to comprehend all of the tasks with one architecture are highly encouraged.

7. Since different random initialization or data order can result in considerably higher scores, we require the bulk submission of at least three results with different random seeds.

8. Every submission is required to have an accompanying description. It is recommended to include the link to the source code.

**Scoring.** To provide an objective means of comparison with the previously published results, we decided to retain the initially formulated metrics.

Regarding the overall score, we consider resorting to an arithmetic mean of different metrics desirable due to its simplicity and straightforward calculation and analysis. Moreover, it is partially justified as all ANLS, F1, and accuracy are interrelated variants of the same measure.

To discount for a different number of tasks within each type, we perform a two-step averaging, i.e., average scores within each category and then average aggregated scores of Document QA, KIE and Table QA groups.[3]

---

[3]Scores on the DocVQA and InfographicsVQA test sets are calculated using the official website.

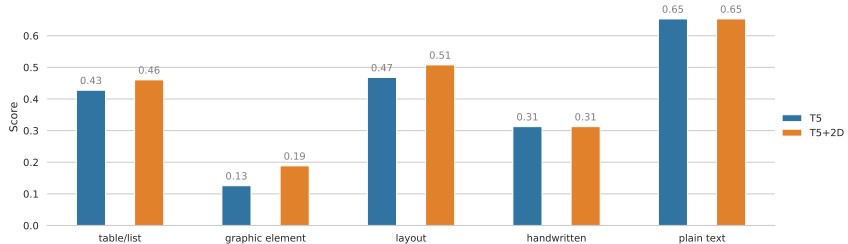

Figure 3: Demonstration of one of the diagnostic subsets we introduced. Here, the impact of 2D bias on cases with particular evidence types can be easily verified.

## 4   Experiments

We intended to facilitate future research by providing extensible model with a straightforward training procedure which can be applied to all of the proposed task in an end-to-end manner. Consequently, we decided to rely on the extended T5 model to ensure these and to identify the current level of performance on the chosen tasks [39]. The introduced modifications is the addition of 2D positional bias [55, 38, 63] that has been shown to perform well on tasks that demand layout understanding. Both models are released with the benchmark on the MIT license. Following the evaluation protocol, the training is run three times for each configuration of model size, architecture, and OCR engine.

Comparison of the best-performing baselines in relation to human performance and top results reported in the literature is presented in Table 3. In several cases there is a significant gap between performance of our baselines and the external best. It can be attributed to several factors. First of all, WTQ and TabFact were reformulated in a document understanding paradigm. External bests for these are no longer applicable to the benchmark in its present, more demanding form. Moreover, they were task-specific, i.e., were explicitly designed for particular task and do not support other datasets within the benchmark. Secondly, there are differences between the evaluation protocol that we assume and what the previous authors assumed (e.g., we do not allow training models on the development sets, we require reporting an average of multiple runs, we disallow pretraining on datasets that might lead to information leak). Thirdly, to simplify the process and ensure easier reproductibility, we did not conduct any unsupervised pretraining (contrary to the most state-of-the-art models). Fourthly, there is no aspect of vision comprehension in our baseline that could possibly address the C3 challenge. Finally, there is the case of Kleister Charity. An encoder-decoder model we relied on as a one-to-fit-all baseline cannot process an entire document due to memory limitations. As a result, the score was lower as we consumed only a part of the document.

Irrespective of the task and whether our competitive baselines or external results are considered, there is still a large gap to humans, which is desired for novel baselines. Moreover, one can notice that the addition of 2D positional bias to the T5 architecture leads to better scores, which is yet another result we anticipated as it suggests that considered tasks have an essential component of layout comprehension. Availability of the diagnostic subsets we introduced allows one to verify this assumption. Figure 3 compares baselines w.r.t. the evidence form and shows that the difference can be attributed to a better comprehension of tables, lists, layout, and even graphic elements where spatial relationships play a pivotal role. We hope the research community will use these and other diagnostic subsets to investigate particular approaches' trade-offs, locate current bottlenecks and answer the question of where should we look for improvement?

## 5   Relation to existing benchmarks and evaluation campaigns

The benchmarks in existence consider either well-established NLP tasks in separation (e.g., Question Answering, Language Modeling, or Natural Language Inference) or a particular aspect of models applied in the field, such as performance w.r.t. the input sequence length. Consequently, recurring evaluation campaigns in Document Understanding can be considered to be related works despite being distinct from a benchmark *per se*.

**NLP benchmarks.** The most recognizable NLP benchmarks of GLUE and SuperGLUE cover a wide range of problems related to language understanding, such as semantic similarity and Natural Language Inference [52, 51]. In contrast to the present work, they are related to short text excerpts lacking layout and accompanying graphical materials. Longer documents and documents collections are covered by decaNLP casting a variety tasks as question answering over a context [30], KILT assuming comprehension grounded in real-world knowledge [37], or Long Range Arena focused on the computational efficiency of the models [48]. All of the mentioned consider plain-text documents without the rich structure that we are aiming at.

Recently, there is a growing interest in dynamic benchmarks enabling customizable model comparison or with tasks changing through time [33, 58, 27]. Our approach is to some extent related as we focus on customization, e.g., multiple leaderboards are available, and it is up to the participant to decide whether to evaluate the model on an entire benchmark or particular category. Secondly, we place attention on the explanation by providing means to analyze the performance concerning document or problem types (e.g., using the diagnostic sets we provide). Finally, we intend to gather datasets not included in the present version of the benchmark to facilitate evaluations in an entire field of Document Understanding, regardless of if they are included in the current version of the leaderboard.

**Evaluation campaigns.** So far, efforts of the Document Understanding community were focused on recurring shared tasks collocated with the major conferences in the field. The most prominent of them is the Robust Reading Competition collocated with the ICDAR. Though not all of the RRC fit into the Document Understanding as we define it, this is where tasks involving information extraction from historical handwritten records and receipts, text extraction from biomedical figures, and document visual question answering have been proposed [29, 18, 57, 12]. An essential difference between the recurring events and a benchmark is that tasks from historical competition require the re-assessment of difficulty in spite of the current state-of-the-art. Additionally, they are often considered in separation, while the benchmark intends to measure system abilities on various tasks at once.

## 6  Conclusions

To efficiently pass information to the reader, writers often assume that structured forms such as tables, graphs, or infographics are more accessible than sequential text due to human visual perception and our ability to understand a text's spatial surroundings. We investigate the problem of correctly measuring the progress of models able to comprehend such complex documents and propose a benchmark – a suite of tasks that balance factors such as quality of a document, importance of layout information, type and source of documents, task goal, and the potential usability in modern applications.

We aim to track the future progress on them with the website prepared for transparent verification and analysis of the results. The former is facilitated by the diagnostics subsets we derived to measure vital features of the Document Understanding systems. Finally, we provide a set of solid baselines, datasets in the unified format, and released source code to bootstrap the research on the topic.

| Dataset / Task type | Score (task-specific metric) | | | |
| --- | --- | --- | --- | --- |
| | T5 | T5+2D | External best | Human |
| DocVQA | 72.5 | 74.1 | 87.1 [38] | 98.1 |
| InfographicsVQA | 37.8 | 43.1 | 61.2 [38] | 98.0 |
| Kleister Charity | 57.9 | 57.7 | 83.6 [63] | 97.5 |
| PWC★ | 24.2 | 25.2 | — | 51.1 |
| DeepForm★ | 73.4 | 74.8 | — | 98.5 |
| WikiTableQuestions★ | 32.5 | 33.4 | 51.8 [60] | 76.7 |
| TabFact★ | 52.2 | 53.7 | 83.9 [10] | 92.1 |
| Visual QA | 55.2 | 58.6 | — | 98.1 |
| KIE | 51.8 | 52.6 | — | 82.4 |
| Table QA/NLI | 42.4 | 43.6 | — | 84.4 |
| Overall | 49.8 | 51.6 | — | 88.3 |

Table 3: Best results of the T5+2D model in relation to human performance and external best.

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
