# OpenReview forum: "Measuring the State of Document Understanding"
_NeurIPS.cc/2021/Track/Datasets_and_Benchmarks/Round1 — Submitted to NeurIPS 2021 Datasets and Benchmarks Track (Round 1)_

### Official Review · Reviewer_uq9V · 2021-07-01
**The paper provides a unified benchmark for a variety of document understanding tasks, an evaluation protocol, and an initial set of baselines for the new benchmark suite.**

**Rating:** 8
**Confidence:** 3

**Strengths:**

Significance of contribution: potentially high. This paper meets calls from the community to iterate on benchmarks. To my understanding, the authors consider a comprehensive set of document understanding tasks.

Relevance to broader research community: likely high. I expect those working on document understanding subtasks will find this useful, but this benchmark may be most helpful to researchers seeking to test their models in the document understanding space.

Societal implications: No concerns. The authors note that the development of this benchmark is unlikely to have negative societal impacts. Having read the paper, I'm inclined to agree. There may be an argument that document understanding will have n-th order impacts on information workers more broadly, but this benchmark paper does not involve any choices that specifically interact with such impacts (as opposed to, e.g. the release of very large document understanding model).

**Weaknesses:**

Significance of contribution: the contribution is ultimately a collection of other tasks, so research focused on specific tasks may not find the aggregation as useful (e.g. researchers focusing on a particular sub-task may not use this benchmark).

**Additional Feedback:**

Very minor:

-I think in-text references to Table 4 should be to Table 3.

-Line 667 "numer"

**Clarity:**

The paper is well written and organized in way that contributions are very clear to readers, as as the Appendix.

**Correctness:**

The datasets provided primarily build on prior work. The modifications made here seem reasonable.  Supplementary materials are also very helpful for clarifying details.

The evaluation protocol seems sound. The choice to withhold the "true" test set could be very useful for this research community (though concerns with creating an excessively leaderboard-focused research culture may be worth discussing).



**Documentation:**

-Augmentations to past datasets are documented in an Appendix.

-Motivation behind including (or not) certain datasets is clear.

-The authors have created a web page for hosting leaderboards, instructions, etc. This looks like it could be a very useful contribution to the community as well. The full version has yet to be released.

-The paper could include a more explicit statement about plans for hosting, licensing, and maintenance (e.g. will a "dump" be published via 3rd-party servers).

**Ethics:**

As noted above, I think the authors are fairly safe to claim this particular work does not have direct negative societal impacts.

**Relation To Prior Work:**

-Very good coverage of the broad document understanding research space. Sections 2 and 3.2 will be useful in their own right.

-Coverage of, and comparison with, past benchmarks.

-Critically, the paper spends a decent of space motivating the document understanding research space. This is likely to increase the overall impact of the submission.

**Summary And Contributions:**

[Updated July 19 2021]
Thanks to the authors for their engagement with the reviews. After the round of discussion, I feel more confident in arguing for acceptance of this paper. In particular, the increased documentation addresses some of the feedback. The sub-task addition will also be useful.

----


In this paper, the authors describe how they collected and augmented a variety of datasets used to study tasks under the umbrella of "document understanding". The paper describes how these tasks cover different aspects of document understanding and provides a protocol for standardized benchmarking in this domain. The contribution of this paper includes both the suite of datasets and tasks (prepared in unified format, with consistent evaluation standards, etc.) and summary of the importance of document understanding and current state of the field. The authors also provide an initial set of experimental results as baselines for other researchers, although this is not the focus of the paper.

---

### Official Review · Reviewer_8jTw · 2021-07-03
**The paper summarises a series of criteria for the dataset of Document Understanding and reformulates datasets from several existing tasks (e.g., Visual Question Answering) based on the proposed criteria.**

**Rating:** 5
**Confidence:** 4

**Strengths:**

1. To unify the dataset of Document Understanding, the paper summarises a series of criteria w.r.t. data gathering, examining, unifying, evaluating, and opening.
2. Based on the summarised criteria, the authors reformulate some existing datasets and obtain their uniform formats.


**Weaknesses:**

** Contribution
The main contribution of this paper is unclear. Summarise some criteria for datasets? Or reformulate the existing datasets? However, no matter which one, the contributions of this paper are limited.

** Presentation
1. The organisation of this paper is messy. For example, in the Introduction, the authors describe the challenges of Document Understanding in a considerable space, which, however, are not verified or discussed in the following sections.
2. The analysis of reformulated datasets is not enough. If the paper focuses on reformulating and unifying the existing datasets, more detailed discussions are required.

** Experiment
1. The experiments are not sufficient, e.g., in the experiment, it considers a single baseline method (i.e., T5) only.
2. The motivation of Figure 3 is not clear. Specifically, Figure 3 can only show that T5 with 2D positional bias (T5+2D) is better than the original T5. It seems the results are irrelevant with the contribution (i.e., dataset) at all.
3. In Table 3, compared with “external best”, why the results of the baseline T5 deteriorate dramatically. Besides, I also have no idea what Table 3 tries to verify.


**Additional Feedback:**

N/A

**Clarity:**

The organisation of this paper is messy, and some claims are a lack of justification, e.g., the discussions of challenges without any proof. Thus, I suggest refining the paper thoroughly, which may make it more logical and readable.

**Correctness:**

The claims in this paper may be partially correct. Specifically, this paper seems to create datasets, but in practice, it collects and reformulates some existing datasets based on several criteria (listed by authors). The analysis and experiments are insufficient as well.

**Documentation:**

The details are partially provided. Specifically, the authors describe the format of data organisation, but lack the other aspects like maintenance, ethics, responsibility, etc.

**Ethics:**

No ethics has been discussed in this paper. One of my main concerns is whether the reformulated datasets would occur ethical issues, as all of them are based on existing datasets.

**Relation To Prior Work:**

The paper does not mention the difference between this work and previous ones.

**Summary And Contributions:**

The authors review the importance and challenges of Document Understanding (DU) and then list a series of criteria (w.r.t. gather, examining, etc) the DU dataset should be satisfied. After that, they consider several tasks (e.g., Question Answering), which can be regarded as branches of DU, and reformulate and unify the datasets of the considered tasks according to the listed criteria above.

---

### Official Review · Reviewer_2S9y · 2021-07-04
**Interesting benchmark but writing/contributions lacking clarity**

**Rating:** 6
**Confidence:** 2

**Strengths:**

The challenges the benchmark seeks to address are significant. A significant amount of information is present as semi-structured data in tables and graphics. Developing better machine learning tools to extract these types of information would have many valuable applications.
Though work on these problems is not new, the authors suggest that there are several challenges with the existing evaluation regime.
- A lack of a consensus on what constitutes a document understanding task. Specifically, the authors contend that many different tasks--from key information extraction to figure QA--fall under the umbrella of “document understanding.” As these tasks involve similar types of reasoning, it makes sense to analyze performance on these datasets together.
- Different architectures are frequently evaluated with different OCR engines. This can confound results, as the choice of OCR engine may have a larger impact than the choice of architecture
- Different formats for datasets, making it difficult to evaluate a model cohesively across many datasets at once.
- A lack of understanding on the precise origin of errors.
- A lack of understanding on what “human performance” is for various datasets.

In response to these challenges, the authors do the following.
1. The authors provide a nice overview of existing tasks that plausibly fit under the umbrella of document understanding. They justify their rationale for the datasets they select to include in the benchmark.
2. The authors standardize to a single OCR engine, enabling better uniformity in results.
3. The authors do important legwork in normalizing data formats for ease of use.
4. The authors identify meaningful auxiliary validation subsets that correspond to important subsets of the datasets. These diagnostic subsets will be useful for comparing model performance at a finer notion of granularity, and understanding the specific tradeoffs that different architectures present.
5. Finally, the authors undertake the effort to derive human performance for several of these benchmarks. They do so in a principled way.


**Weaknesses:**

1. The authors could do a better job explaining why they chose to ignore certain tasks in the construction of the benchmark--especially after spending time discussing them in the paper. For instance, the benchmark contains no classification task.


2. It would be helpful to understand why the task of richly formatted KBC is not included in the benchmark? Ostensibly, it implicates many of the same problems that exist in the other datasets that are included. My understanding is that because KBC focuses on converting tables to list of triples, it doesn’t process a document as a cohesive unit (in the same way as the other tasks that are included). Still, more discussion would be helpful.

3. One cause for concern is that the model evaluated (T5) performs significantly worse than the external state-of-the-art for each of the publicly available datasets. This would raise several questions. Why didn’t the authors try to evaluate/adapt any of the models featured in the “external best” column of Table 3? If any of those models don’t adapt to the other tasks, than some discussion of why that is would be useful. At present, there is a danger that the performance of the author’s chosen model cuts against the broader thesis of treating these datasets as a common benchmark. Because it is so poor relative to other models, a reader may question the virtue of considering these datasets jointly.

4. It would be helpful if the authors could explain more on _why_ these 7 datasets should be considered together. What is it about these tasks that fundamentally suggests they should be treated together? The authors allude to the idea that in all the these tasks, the model must comprehend the "layout" of the documents. A bit more precision and exposition here would be helpful.


**Additional Feedback:**

I think the challenges raised by this benchmark are interesting. The authors should focus on improving the quality of the writing and being clearer about their contributions: what is the current gap in the field that this work fills? Why do we need this benchmark, when many of the datasets themselves already exist?

**Clarity:**

No. The paper is quite difficult to read, and it is challenging to follow the author’s arguments or discern their exact contributions.

The authors spend significant time discussing the virtues of a benchmark: quality, difficulty, licensing. These seem like straightforward qualities that are widely accepted? Perhaps the authors would be off spending their time discussing the specific qualities of a rich formatting document QA benchmark.

The authors need to be more precise about their contributions. Given that many of the datasets included were created by other parties, it would be good to be clearer about the novel contributions that this work itself presents. It takes multiple readings to discern what the contributions are. Even then, I'm a bit unsure. A clearly bulleted list at the end of the introduction for example, would greatly improve the paper.

The authors would benefit from providing examples when discussing each task. It is difficult to follow the discussion distinguishing different tasks without concrete examples of each.

I think the paper would greatly benefit from some type of table, comparing each of the types of tasks, the structure of their typical inputs/outputs, and an example of the types of reasoning they require.

The authors’ use of capitalization is questionable -- I don’t think the task types (“document understanding”) should be capitalized.

**Correctness:**

The claims seem broadly correct. The evaluation methods are sound. The choice to identify diagnostic subsets is especially nice, given the increasing focus across the field on more granular forms of evaluation.

**Documentation:**

Largely, yes.

Nit: Some of the pages on the benchmark website don’t work. When I clicked on “Analysis” under https://duebenchmark.com/leaderboard/key-information-extraction, I got a 404 error.

**Ethics:**

There don’t appear to be any obvious ethical concerns arising from this work.

**Relation To Prior Work:**

Somewhat. The authors provide a nice overview of tasks falling under the umbrella of “document understanding.” This is helpful and valuable.

However, the authors fail to sufficiently situate their work in the context of prior work. This is in part difficult because--at times--it is hard to discern the exact contributions relative to prior work. The authors don’t sufficiently clarify whether there are (1) no existing richly formatted document understanding benchmarks, or (2) whether the existing ones are inadequate in some way.

Perhaps an analogy to similar efforts for other tasks/domains would be helpful. The most obvious analogy seems to be Glue/SuperGlue: here--as in there--the authors are unifying a set of disparate tasks and datasets under a common benchmark, because they believe that these tasks implicate a common set of ML challenges.


**Summary And Contributions:**

The paper introduces a benchmark for document understanding in rich layout settings (i.e. when information is contained in tables, graphs, lists, and infographics). The benchmark encompasses 7 datasets involving question-answering, entailment, and information extraction. Though the majority of the datasets themselves previously existed, the authors take steps to clean errors, normalize input/output, and modify the task format to map the broader goals of the benchmark. The authors additionally identify “validation subsets” corresponding to meaningfully distinct dataset slices that researchers might hold specific interest in. Finally, the authors present empirical results for existing methods.

---

### Decision · Program_Chairs · 2021-07-27

**Decision:**

Reject

**Comment:**

The paper presents an aggregated, unified benchmark related to document understanding which repurposes many previously published benchmarks (i.e., WikiTableQuestions, PWC (Papers with Code), DeepForm) focusing on extracting semantics from the original document form (PDF). Along with this, it presents baselines and leaderboards, along with an OCR engine that helps comparing different models while not accounting for OCR performances, and new, in-house human annotation to estimate human performance.

Unfortunately, it is not well justified why such such "document understanding" benchmark is needed (pointed by Reviewer 2S9y, 8jTw), and what new research directions it would support for the community. The experiments should provide in-depth analysis on how to document layout understanding impacts performances on this benchmark, in its current form, it is unclear what is the main bottlenecks on this dataset. i.e., how do the challenges listed in C1-C4 affect the performances? Two simple baselines are presented without in-depth discussions, and some design decisions are not well motivated or justified (e.g., they document the performance of their baseline is low because they did not conduct any unsupervised pre-training, but T5 comes with a large amount of unsupervised pretaining).

The appendix provides ample details of data collection, but I find the writing and organization of the paper confusing (e.g., what is section 1.3 "Desiderata" capturing?), and some crucial information (e.g., inter-annotator agreement for human evaluation) missing.

Lastly, one relevant reference: https://arxiv.org/pdf/2106.00676.pdf